# Evaluating the appropriateness of laboratory testing and antimicrobial use in South African children hospitalized for community-acquired infections

Lydia Mapala [ORCID] *☯, Adrie Bekker☯, Angela Dramowski☯

Department of Paediatrics and Child Health, Faculty of Medicine and Health Sciences, Stellenbosch University and Tygerberg Hospital, Cape Town, South Africa

☯ These authors contributed equally to this work.
* lydiamapala@gmail.com

## Abstract

### Introduction

Community acquired infection (CAI) is the leading indication for paediatric hospitalization in South Africa.

### Methods

We conducted secondary data analysis of prospective, consecutive paediatric admissions to Tygerberg Hospital (May 2015-November 2015). Clinical characteristics, admission diagnosis, appropriateness of diagnostic tests, use of antimicrobial prescriptions, hospital outcome and costs were analyzed.

### Results

CAI episodes were documented in (364/451; 81%) children admitted to the general paediatric ward; median age 4.8 months (Interquartile range, IQR, 1.5–17.5) and weight 5.4kg (IQR, 3.6–9.0). Male gender predominated (210/364; 58%), and Human Immunodeficiency Virus infection prevalence was 6.0% (22/364). Common CAI types included respiratory tract infections (197; 54%), gastroenteritis (51; 14%), and bloodstream infections (33; 9%). Prehospital antibiotics (ceftriaxone) were given to 152/364 (42%). Of 274 blood cultures and 140 cerebrospinal fluid samples submitted, 5% and 2% respectively yielded a pathogen. Common CAI antibiotic treatment regimens included: ampicillin alone (53%); ampicillin plus gentamicin (25%) and ampicillin plus cefotaxime (20%). Respiratory syncytial virus (RSV) was found in 39% of the children investigated for pneumonia. Most antibiotic prescriptions (323/364; 89%) complied with national guidelines and were appropriately adjusted based on the patient's clinical condition and laboratory findings. The overall estimated cost of CAI episode management ZAR 22,535 ($\approx$1423 USD) per CAI admission episode. Unfavourable outcomes were uncommon (1% died, 4% required re-admission within 30 days of discharge).

**Data Availability Statement:** All relevant data are within the paper and its Supporting Information files.

**Funding:** AD is supported by a NIH Fogarty Emerging Global Leader Award K43 TW010682 and received funding for the laboratory work from the SAMRC through a Self-initiated Research (SIR) Grant. The funders had no role in study design, data collection and analysis, decision to publish, or preparation of the manuscript.

**Competing interests:** The authors have declared that no competing interests exist.

## Conclusion

CAI is the most frequent reason for hospitalization and drives antimicrobial use. Improved diagnostic stewardship is needed to prevent inappropriate antimicrobial prescriptions. Clinical outcome of paediatric CAI episodes was generally favourable.

## Introduction

Community acquired infection (CAI) is the leading indication for hospitalization in general paediatric wards in low and middle-income countries (LMICs) [1]. CAIs include all infections arising outside of the healthcare setting that are already present or incubating at the time of hospital admission. Common types of CAI include severe bacterial infections (bloodstream infection, pneumonia, meningitis, urinary tract infection, skin and soft tissue infection), gastroenteritis, respiratory virus infections, viral infection syndromes (e.g., herpes stomatitis, and varicella) and fungal infections.

Community-acquired pneumonia is the leading cause of child morbidity and mortality worldwide, with an incidence in LMICs of 0.22 episodes per child-year [2]. The main risk factors for development of CAP are young age, malnutrition, low birth weight and Human Immunodeficiency Virus (HIV) infection. Globally there has been a 30% decrease in the incidence of CAP from 2010 to 2015, mostly attributed to prevention of vertical HIV transmission and improved pneumococcal conjugate vaccine (PCV) uptake [3]. *S. pneumoniae* remains the most important bacterial pathogen, although the incidence has declined significantly following introduction of the PCV-7 and more recently PCV-13 [4]. Owing to the difficulty in obtaining bacteriological identification in cases of childhood CAP, antibiotic treatment is usually presumptive.

Respiratory Syncytial virus (RSV) is the most common viral pathogen in children and is especially prevalent in the winter months in temperate climates. In the Drakenstein study in South Africa, RSV was the most common viral pathogen identified in hospitalized children with respiratory illness, although the bacterium *B. pertussis* was also frequently isolated. Between 2011–2016 24.1% and 5.5% of 3645 hospitalized children aged of 0–59 months, tested positive for RSV and influenza virus infections, respectively. The large overlap in the clinical, laboratory and radiological findings in bacterial versus viral CAP, has led to recommendations for routine antibiotic use in young children with severe CAP [5, 6].

The World Health Organization (WHO) developed the Integrated Management of Childhood Illness guidelines (IMCI) for use in primary care facilities to assist with triage of ill-children based on the severity of their presenting symptoms and signs (S1 Appendix). For acutely ill children requiring transfer from primary care to hospital, the IMCI guidelines and the South African Essential Drug List (EDL) recommends a pre-transfer dose of intravenous or intramuscular third-generation cephalosporin e.g., ceftriaxone. Oral amoxicillin is first line therapy for non-severe pneumonia. Ampicillin plus gentamicin is recommended for infants less than 3 months of age with severe pneumonia and ceftriaxone for older children with severe pneumonia (S2 Appendix). Oral amoxicillin is the first line therapy for non-severe pneumonia as it provides adequate cover for the most common CAP pathogens (*Streptococcus pneumoniae* and *Haemophilus influenzae)*. However, amoxicillin has little activity against enteric Gram negative bacteria e.g. salmonella, that are important causes of paediatric CAP in some parts of Africa [7].

Gastroenteritis (GE) is the second most common cause of childhood morbidity and mortality globally with an estimated 1.7 billion cases of diarrhoea in 2010 [8]. While rotavirus is the most prevalent GE pathogen worldwide, the incidence of rotaviral-associated GE death and

hospitalization rate has substantially declined in South Africa and the continent since inclusion of the rotavirus vaccine in the national immunization schedule in 2009 [9–11]. Antibiotics are not routinely recommended for diarrhoeal disease as the causative pathogens are mostly viruses. Antibiotics are used in certain situations such as severe malnutrition and for infants less than 28 days (ampicillin and gentamicin). In cases of suspected dysentery, ceftriaxone or ciprofloxacin is the EDL- recommended treatment.

Febrile illnesses are a common presentation in young children requiring hospital admission. Severe bacterial infections including bloodstream infection (BSI) or bacteremia, meningitis, and urinary tract infections, typically present as a febrile illness in infants and young children. For this reason, most febrile infants undergo blood culture sampling, urine screening and in some cases, a lumbar puncture on hospital admission. BSI may occur in isolation (so-called primary BSI) or may be secondary to infection at another site e.g., *S. pneumoniae* BSI with community-acquired pneumonia (CAP) or meningitis, or *E. coli* BSI arising from a urinary tract infection (UTI). A study profiling trends in paediatric BSI at Tygerberg Hospital (2008–2013) showed that Gram negative infections predominated, with *E. coli* and *K. pneumoniae* being the most CA-BSI common pathogens [12].

Meningitis is another common indication for laboratory investigation and hospitalization in children. The diagnosis of meningitis in children relies on clinical suspicion and the correct interpretation of appropriate laboratory test results, if available. Aseptic meningitis remains the predominant type of meningitis reported in the literature [13] although this could partly be attributed to the administration of pre-hospital transfer antibiotics given according to the IMCI guidelines. Empiric treatment of suspected meningitis is ampicillin plus cefotaxime in infants < 3 months of age, with ceftriaxone alone used for older infants and children. Urinary tract Infections are most common in the first 6 months of life. Symptoms of UTI are often non-specific, especially in infants, therefore it is important to exclude the diagnosis in all febrile children. Diagnosis is made on urine specimens taken aseptically. Risk factors for development of UTI include anatomical urinary tract abnormalities and female gender. Common UTI pathogens include the Enterobacteriaceae family with *E.coli* as the most common pathogen overall (70–90% of cases). *Staphylococcus aureus* and *Candida albicans* are other important pathogens [14]. For non-severe urinary tract infections in infants and children > 3 months old, the recommended therapy is oral amoxicillin-clavulanic acid. Intravenous amoxicillin-clavulanic acid or ceftriaxone was the EDL-recommended treatment for severe UTI and UTI in infants <3 months of age.

In high-income settings, laboratory investigations are routinely submitted to identify the causative CAI pathogen/s (bacterial, viral, fungal or other), however in many parts of Africa, microbiological diagnostic services are unavailable and the diagnosis of CAI is made clinically. Conducting laboratory tests may assist with clinical decision-making including the need for, type and duration of antimicrobial therapy. Although some data exists on certain CAI pathogens in South African children, there is limited information on the CAI burden, clinical management and outcome of hospitalized children. In addition, the recent introduction of several key vaccines targeting CAI pathogens (rotavirus, pneumococcal conjugate vaccine) has changed the profile of vaccine preventable diseases and paediatric hospitalizations. Understanding the burden of CAI and the profile of causative pathogens may assist with developing guidelines for appropriate investigation and antimicrobial management, ultimately contributing to improved childhood health outcomes. In this study we describe the burden, spectrum, pathogen profile, clinical management and outcome of children hospitalized for CAI in a general paediatric ward. In addition, we analyze the cost and appropriateness of diagnostic laboratory testing performed and empiric antimicrobial therapy prescribed for CAI in a cohort of hospitalized South African children.

## Methods

### Study setting

The study was undertaken at Tygerberg Hospital, a tertiary hospital in the Western Cape Province of South Africa. The hospital has a 1384-bed capacity including 300 paediatric and neonatal beds (0–13 years). The inpatient admissions in 2015 were 17140, 11238 (65.6%) to the general paediatric service and the remainder to medical specialty paediatric wards, paediatric surgery and the paediatric intensive care unit (PICU). The general paediatric ward has a 25-bed capacity, with 88% occupancy and an average length of stay of 6.5 days in 2015. The ward admits children from surrounding primary care clinics, district hospitals and the paediatric short-stay ward, and provides step-down care of patients discharged from the PICU.

### Study design

We conducted secondary analysis of data from a prospective cohort study of consecutive patient admissions to the general paediatric ward at Tygerberg Hospital between 11 May 2015 and 10 November 2015. The original study conducted daily prospective clinical surveillance of all admitted patients (0–13 years) to detect development of healthcare-associated infections (HAI), with daily review of clinical notes, laboratory investigation results, antimicrobial prescription charts and final outcome of hospital stay [15]. The data base was accessed again from August 2017 for the present study. The present study determined the proportion of all admissions with CAI as the main indication for hospitalization. Patients considered to have a healthcare-associated infection (readmission within <30 days of hospital discharge) and patients with a non-infectious indication for hospitalization were excluded from further analysis. Ethical approval and waiver of individual informed consent was obtained from the Human Health Research Ethics committee of Stellenbosch University (S13/09/171) and study approval was granted by the Tygerberg Hospital management.

### Inclusion criteria

All children 0–13 years admitted to the general paediatric ward over the 6-month (11 May 2015 and 10 November 2015) study period were eligible for inclusion in this study if their primary indication for hospitalization was a CAI episode and the hospitalization episode lasted at least 48 hours. Pathogens isolated from laboratory samples (urine, CSF, blood, stool, respiratory aspirates) collected ≤ 48hrs after admission were classified as CAI, in patients with no prior history of hospitalization in the preceding 30 days.

### Exclusion criteria

Patients considered to have a healthcare-associated infection (readmission within <30 days of hospital discharge) and patients with a non-infectious indication for hospitalization were excluded from further analysis.

### Study definitions and data sources

CAI episodes were classified as described in the hospital records as respiratory tract infections, gastroenteritis, bloodstream infection, meningitis, urinary tract infection and other infections, although some patients presented with ≥1 infection. Pathogens isolated from laboratory samples (urine, CSF, blood, stool, respiratory aspirates) collected ≤ 48hrs after admission were classified as CAI, in patients with no prior history of hospitalization in the preceding 30 days. Following hospital discharge, patient hospital and laboratory records were searched to identify patients who had been re-admitted within 30 days of discharge.

Additional laboratory variables were collected retrospectively from patient records and the National Health Laboratory Service (NHLS) websites (Disalab and LabTRAK) including C-reactive protein (CRP), white blood cell count (WCC), platelet count, and other microbiology and virology laboratory specimens sent to determine the aetiology of the CAI episode. Investigations of patients were noted to be inappropriate if they did not conform to the South African Essential Drug List (EDL) guidelines (S2 Appendix) and the clinical picture described in the hospital record. This included failure to perform blood culture in patients considered to have bloodstream infection and failure to perform lumbar puncture in patients admitted with suspected meningitis. Empiric treatment of CAI at Tygerberg Hospital is guided by the suspected focus of infection and the South African EDL. Therapy for severe CAP includes ampicillin and gentamicin, or ampicillin alone in non-severe pneumonia in a child that is unable to swallow. If meningitis is suspected, cefotaxime plus ampicillin or ceftriaxone plus ampicillin is used for infants, or a 3rd generation cephalosporin alone in older children. Antibiotics for acute diarrheal disease are generally prescribed for specific infections e.g. ciprofloxacin for dysentery, and metronidazole for *G. lamblia* infections. In addition, antibiotics may be added for gastroenteritis for neonates and children with shock or malnutrition. Standard treatment for uncomplicated UTI includes amoxicillin-clavulanic acid orally or IV ceftriaxone for complicated UTI. The IMCI guidelines (S1 Appendix) recommend the administration of a stat dose of antibiotics to all potentially serious bacterial infections and this was taken into account when collecting and interpreting findings as it may have influenced culture yield.

The 2013 version of the South African EDL was used to determine the appropriateness of the empiric antimicrobial prescription. Adherence to principles of antimicrobial stewardship was evaluated e.g. the proportion of prescriptions switched from intravenous to oral agents by day 3, appropriate de-escalation from broad- to narrow-spectrum agents after pathogen identification and discontinuation of antibiotics if a viral pathogen was identified and considered to be the sole cause of infection. Inappropriate antimicrobial use was defined as: use of intravenous medications in patients with non-severe infection and ability to tolerate oral feeds; use of antibiotics in acute diarrhoeal disease (excluding dysentery and Giardiasis); use of intravenous antibiotics for uncomplicated UTI and failure to modify antimicrobial prescription after receipt of culture and sensitivity results.

Cost analysis of the impact of CAI was performed from the healthcare provider perspective using the 2015 costs entered into the formula (for each of the subtypes): number of CAI events × average length of stay for that subtype × unit cost per patient. The unit cost per patient was ZAR 2916, which included all laboratory investigations, radiology and pharmaceutical costs.

### Data management and statistical analysis

Data were entered into an institutional-hosted REDCap database [16]. Descriptive analysis of demographic characteristics was performed reporting continuous variables as median (IQR), and categorical data as proportions or percentages. All the statistical analyses were performed using STATA 16.0 (College Station, Texas 77845 USA).

### Results

A total of 451 children were admitted to the general paediatric ward over the 6-month study period. Of these admissions, 364/451 (81%) were hospitalized with a primary diagnosis of one/more CAIs. The spectrum of CAI episodes included community-acquired pneumonia—CAP (54%; 197), diarrhoeal disease (14%; 51), mixed infections with >1 infection type (12%; 43), bloodstream infection (BSI) (9%; 33), meningitis (5%; 17), UTI (2%; 8), and other infection types (4%; 15)Among children with CAI, the median age was 4.8 months (IQR 1.5–17.5), with

a male predominance and an HIV-infection prevalence of 6.0% (22/364) Severe acute malnutrition (SAM) was present in 29/364 (8%). Underlying medical and surgical conditions were frequent (46/364; 12%), including neurological disorders/cerebral palsy (17; 5%), congenital cardiac lesions (7; 2%), chronic lung disease (7; 2%), chromosomal abnormalities (6; 2%), chronic kidney disease (4; 1%), sickle cell disease (3; 1%) and suspected primary immune deficiency (2; 1%). More than half of the children (225/ 364; 62%) with CAI were transferred in from other health facilities; 139 (38%) came directly from home. Of the referred children, 152 (68%) had received pre-hospital empiric antibiotic therapy. (Among the 22/364 (6%) of children living with HIV, 14 (64%) were already on antiretroviral therapy (ART) but only 6 (43%) had achieved virological suppression (HIV RNA <50 copies/ml); their median CD4 cell count was 451 (IQR: 371–1501). Pneumonia was the most frequent admission diagnosis 15 (68%) among HIV-infected children, and three-quarters (17/22; 75%) had received recent or current treatment for confirmed tuberculosis disease (pulmonary, meningitis or disseminated TB); 15/ 17 (88%) for drug-susceptible TB. (Table 1).

Clinical haematology (White cell count, platelets), chemistry (C-reactive Protein), microbiology and virology laboratory test utilization rates were high with 360/364 (99%) children having had one/more laboratory test/s submitted. A total of 1206 tests were performed; patients who presented with pneumonia accounted for the most diagnostic tests submitted overall (522/1206, 43%). Of 274 blood cultures and 140 cerebrospinal fluid samples submitted, only 5% and 2% respectively yielded a pathogen.

Respiratory syncytial virus (RSV), adenovirus and parainfluenza virus predominated from shell vial culture tests, whereas RSV was the most frequent virus identified on respiratory virus polymerase chain reaction testing (RV PCR). Most children with CAI (303/364; 83%) were considered to be appropriately investigated; only 9/364 (3%) had inappropriate investigations and 52/364 (14%) had no investigations performed (which in most cases was deemed appropriate as they presented with probable viral illnesses).

In 323/364 (89%) patients, empiric antibiotic usage was EDL guideline compliant. Common CAI antibiotic treatment regimens included: ampicillin alone (53%); ampicillin plus gentamicin (25%) and ampicillin plus cefotaxime (20%). For 17/364 (5%) patients, prescriptions were not considered to be guideline compliant. This included administration of antibiotics in children presenting with acute gastroenteritis with no risk of a bacterial infection. Of those on empiric therapy (n = 340), antibiotics were stopped in 169/340 (50%) patients after clinical improvement and negative laboratory tests. De-escalation of antibiotic therapy, such as change from ampicillin plus cefotaxime to ampicillin plus gentamicin was noted in 38/340 (11%) of cases. In 39/340 (12%) children who needed antibiotic escalation, all patients received appropriate escalation based on clinical presentation and laboratory results (Tables 2 and 3).

Outcomes of paediatric hospitalization for CAI were generally favourable, however 3 children (1%) died from *E.coli* sepsis, pneumonia (with underlying cerebral palsy and hydrocephalus) and septic shock (underlying spastic cerebral palsy and disseminated TB). Fourteen children (3.8%) required re-admission to hospital within 30 days of discharge, with age ≤ 3 months and a history of preterm birth as risk factors in this group (Table 1).

In this study population, hospital costs for CAI was estimated to be just over ZAR 8 million. Community acquired pneumonia contributed 49% to the estimated cost followed by diarrheal disease at 15% (Table 4).

## Discussion

In this study, we characterized the epidemiology of CAI in South African children admitted to a general paediatric ward in Cape Town. The patient population was young, predominantly

**Table 1. Characteristics and outcomes of children hospitalized with community-acquired infection (n = 364).**

| | Pneumonia | Diarrhoeal disease | More than one infection type | Bloodstream infection | Meningitis | Urinary tract infection | **Other Infections | Total admissions |
|---|---|---|---|---|---|---|---|---|
| Number, % | 197 (54%) | 51 (14%) | 43 (12%) | 33 (9%) | 17 (5%) | 8 (2%) | 15 (4%) | 364 |
| Sex (male) | 110 | 29 | 29 | 19 | 10 | 4 | 9 | 210 (58%) |
| Median age in months (IQR) | 7 (2.3–23.5) | 13 (1.6–15.5) | 3.3 (0.7–4.1) | 1 (0.7–9.3) | 27.1(2.1–53.6) | 10.9 (8.4–30) | 5.9 (1–16.6) | 13 (1.5–17.5) |
| HIV status | | | | | | | | |
| HIV infected | 15 (8) | 2 (4) | 3 (7) | 2 (6) | 0 | 0 | 0 | 22 (6) |
| HIV-exposed uninfected | 28 (14) | 19 (37) | 4 (9) | 6 (18) | 3 (18) | 0 | 4 (27) | 64 (17) |
| HIV negative | 143 (73) | 30 (59) | 33 (77) | 23 (70) | 14 (83) | 7 (83) | 10 (66) | 259 (71) |
| HIV unknown | 11 (6) | 0 | 3 (7) | 2 (6) | 0 | 1 (13)) | 1 (7) | 18 (5) |
| Premature birth | 40 (20) | 6 (12) | 10 (23) | 8 (24) | 1 (5.8) | 0 | 4 (20) | 69 (19) |
| Weight median (IQR) | 6.2 (4.1–9.4) | 6.6 (3.5–8.3) | 3.9 (3.1–5.0) | 3.7 (2.9–5.30) | 6.4 (4.4–18) | 6.2 (4.7–13.5) | 8.0 (3.5–9.6) | 5.5 (3.6–9.0) |
| Severe acute malnutrition* | 10 (5) | 6 (12) | 1 (2) | 9(27) | 1 (6) | 2 (25) | 0 | 29 (8) |
| Origin of referral | | | | | | | | |
| Home | 82 (42) | 15 (29) | 17 (40) | 14 (42) | 5 (29) | 3 (38) | 3 (20) | 139 (38) |
| Clinic | 56 (28) | 16 (31) | 11 (30) | 13 (39) | 6 (35) | 3 (38) | 7 (47) | 112 (31) |
| Hospital | 59 (30) | 20 (39) | 15 (35) | 6 (18) | 6 (35) | 2 (25) | 5 (33) | 113 (31) |
| Pre-transfer antibiotics | 80 (41) | 26 (51) | 15 (35) | 12 (36) | 9 (53) | 3 (38) | 7 (47) | 152 (42) |
| TB status | | | | | | | | |
| TB treatment | 9 (5) | 1 (2) | 3 (7) | 2 (6) | 1 (6) | 1 (13) | 0 | 17 (5) |
| Not TB | 111 (56) | 36 (71) | 30 (70) | 24 (73) | 12 (71) | 6 (75) | 11 (73) | 230 (64) |
| TB prophylaxis | 4 (2) | 2 (4) | 0 | 0 | 0 | 0 | 0 | 6 (2) |
| WCC on admission (*10^9 cells/litre) median (IQR) | 11 (8–15) | 14 (10–21) | 11 (7.8–18) | 12.0 (9.5–15) | 12 (7.8–15.2) | 12 (10.5–16.5) | 17 (8–17) | |
| C-reactive protein on admission* (mg/L) | 4.7 (4–39) | 24 (4–56.5) | 18.6 (4–85) | 31 (4–100) | 9.6 (4–42.9) | 45.5(63.5–140) | 104 (6–146) | |
| Most frequent empiric antibiotic regimen | ampicillin + gentamicin | ceftriaxone | ampicillin + cefotaxime | ampicillin + cefotaxime | ceftriaxone | amoxicillin clavulinic acid | ampicillin + cefotaxime | |
| Proportion on regimen | 73 (37) | 21 (41) | 19 (44) | 16 (48) | 13 (76) | 3 (37) | 3 (20) | |
| Patient outcome | | | | | | | | |
| Discharged | 177 (90) | 45 (88) | 34 (79) | 27 (81) | 11 (65) | 8 (100) | 11 (73) | 302 (83) |
| Transferred | 19 (9) | 6 (12) | 8 (19) | 6 (19) | 5 (29) | 0 | 4 (27) | 48 (13) |
| Died | 1 (1) | 0 | 1 (2) | 0 | 1 (6) | 0 | 0 | 3 (1) |
| Readmitted | 8 (4) | 0 | 1 (2) | 2 (6) | 0 | 3 (38) | 0 | 14 (4) |

Number and %, unless stated otherwise

*Severe Acute Malnutrition as defined by weight for height <-3 Z score or MUAC <11.5cm or presence of bilateral pedal oedema; IQR = Interquartile range;

TB = Tuberculosis; HIV = Human immunodeficiency virus. WCC = White Cell Count *prematurity; born before 37 completed weeks of pregnancy

**Other infections = (15) viral exanthem (4), eczema herpeticum (2), herpes stomatitis (1), cellulitis (1), septic arthritis (1), fungal skin infections (2), upper respiratory infection (1), conjunctivitis (1), tonsillitis (1), staphylococcal scalded skin syndrome (1).

male (in keeping with published data on paediatric CAI patient demographics) and had a relatively low HIV prevalence rate when compared to previous SA hospital-based studies [12, 17]. Patients with CAI comprised 81% of all ward admissions, highlighting the heavy burden of infectious diseases in paediatric practice in South Africa. Common underlying factors in children hospitalized for CAI were prematurity, malnutrition, and young age.

**Table 2. Laboratory investigations and pathogen yield children with community-acquired infection.**

| Microbiological investigations | | | | |
|---|---|---|---|---|
| Test | Total sent | Positivity rate n, % | Pathogen yield, n, % | Contamination rate, n, % |
| Blood culture MC&S[1] | 274 | 35 (13) | 13 (5) | 22 (8) |
| Urine MC&S[2] | 106 | 30 (28) | 26 (25) | 4 (4) |
| CSF MC&S[3] | 140 | 4 (3) | 3 (2) | 1 (1) |
| Stool MC&S | 20 | 0 | 0 | 0 |
| **Virological investigations** | | | | |
| Test | Total sent | Positive yield n, % | Identified Viruses | Number positive |
| Respiratory virus shell vial culture[4] | 59 | 18 (31) | *See footnote | |
| Respiratory virus PCR[4] | 16 | 11 (69) | | |
| Cerebrospinal fluid PCR[5] | 28 | 2 (7) | Enterovirus | 2 (100) |
| Rotavirus and adenovirus antigen in stool | 19 | 1 (5) | | |

MC&S = microscopy, culture and susceptibility; CSF = Cerebrospinal fluid, PCR = polymerase chain reaction

[1]Blood culture pathogens (n = 13); *E. coli* (6), *Salmonella non-typhi* (2), *S. pneumoniae* (1), *S. thoracolentis* (1), *S. aureu*s (1), *S. agalactiae* (1), *C. meningosepticum* (1)

[2]Urine culture pathogens (n = 26): *E. coli* (13), *K. pneumoniae* (8), *E. faecalis* (1), *C. albicans* (4)

[3]CSF culture pathogens (n = 3): *E. coli* (1), *S. pneumoniae* (1), *S. agalactiae* (1)

[4]Respiratory pathogens: Respiratory Syncytial Virus, (16; 39%), Cytomegalovirus (8; 20%), Adenovirus (6; 15%), Parainfluenza virus type 3 (5; 12%), Human rhinovirus (3; 7%), Influenza (1; 2%), Coronavirus (1; 2%), Human Metapneumovirus (1; 2%)

[5]CSF viral pathogens: Enterovirus (2)

Although a broad spectrum of CAI types was documented, pneumonia and GE remained the leading causes of paediatric hospitalization. Only one child with CAP had a confirmed *S. pneumoniae* infection. The steady decline in cases of invasive pneumococcal disease in South African children can be attributed to the introduction of the PCV vaccine, increased uptake of paediatric antiretroviral therapy and effective prevention of mother to child HIV transmission (PMTCT) [4, 18]. Prematurity was the most frequent underlying factor in children admitted with CAP. RSV was the most frequently isolated viral pathogen followed by adenovirus and CMV, as has been shown in previous studies [2, 5]. Protection of this vulnerable population from RSV infection is difficult, with Palivizumab immunoprophylaxis beyond the means of public sector patients in South Africa. Introduction of a RSV vaccine in future is likely to have a dramatic impact on the profile of paediatric admissions with CAP, particularly for ex-premature infants.

Although rotavirus is the leading cause of GE globally, it was not isolated on stool from any patient in this cohort, possibly owing to the small number of stool specimens collected. Previous studies have highlighted a substantial decline in rotavirus GE associated hospitalization and mortality following introduction of the rotavirus vaccine to the SA-EPI in 2009 [9].

Aseptic meningitis remains the most common type of meningitis encountered in children, highlighted by the very low yield of CSF in this cohort (2%). Failure to isolate bacterial pathogens as a result of antibiotic usage prior to CSF sampling could not be excluded. Although the CSF yield was low, submission of specimens for bacterial culture and viral PCR is useful to identify viral causes of meningitis, where antibiotics can be safely discontinued.

The pathogen profile of UTIs in paediatric populations in South Africa is not well-described. Our study identified *E.coli* and *K. pneumoniae as* the most frequent uropathogens with many being ESBL-producers (38%). Of the patients treated for UTI, 50% received 3rd generation cephalosporins and only 3% received oral amoxillin-clavulanic acid. Regular review of empiric antibiotic guidelines for CA-UTI is needed considering the high rates of resistant pathogens encountered, even among CAI bacterial pathogens.

**Table 3. Infection diagnostics and antimicrobial therapy appropriateness and guideline compliance.**

| Infection type | Community-acquired pneumonia n = 197 | Diarrhoeal disease n = 51 | Mixed infection syndromes n = 43 | Bloodstream infection n = 33 | Meningitis n = 17 | Urinary tract infection n = 8 | Other* n = 15 | Total CAI** n = 364 |
|---|---|---|---|---|---|---|---|---|
| **Laboratory investigations** | | | | | | | | |
| Appropriate | 156 (79%) | 40 (78%) | 42 (98%) | 32 (97%) | 14(82%) | 8 (100%) | 11 (73%) | 303 (83%) |
| Inappropriate | 5 (3%) | 1 (2%) | 0 | 0 (0%) | 3(18%) | 0 | 0 | 9 (3%) |
| No specimen sent | 36 (18%) | 10 (20%) | 1 (2%) | 1 (3%) | 0 | 0 | 4 (27%) | 52 (14%) |
| **Empiric antimicrobial therapy** | | | | | | | | |
| Guideline compliant | 167 (84%) | 43 (84) | 43 (100%) | 32 (97%) | 17 (100%) | 8 (100%) | 13 (87%) | 323 (89%) |
| Not guideline compliant | 13 (7%) | 1 (2%) | 0 | 1 (3%) | 0 | 0 | 2 (13%) | 17 (5%) |
| No antimicrobial given | 17 (9%) | 7 (14%) | 0 | 0 | 0 | 0 | 0 | 24 (6%) |
| **Empiric antimicrobial therapy** | | | | | | | | |
| Appropriately adjusted | 177 (90%) | 44 (86%) | 43 (100%) | 33 (100%) | 17(100%) | 8 (100%) | 15 (100%) | 337 (92%) |
| Not adjusted | 3 (1%) | 0 | 0 | 0 | 0 | 0 | 0 | 3 (1%) |
| No antibiotics given | 17 (9%) | 7 (14%) | 0 | 0 | 0 | 0 | 0 | 24 (7) |
| **Antimicrobial adjustments** | | | | | | | | |
| Antimicrobial/s discontinued | 99 (56%) | 23 (52%) | 12 (28%) | 12 (37%) | 11 (65%) | 4 (50%) | 8 (53%) | 169 (50%) |
| IV to oral switch | 58 (33%) | 14 (32%) | 7 (16%) | 8 (24%) | 0 | 1 (12.5%) | 3 (20%) | 91 (27%) |
| De-escalation | 13 (7%) | 4 (9%) | 14 (33%) | 5 (15%) | 1 (6%) | 1 (12.5%) | 0 | 38 (11%) |
| Appropriate escalation | 7 (4%) | 3 (7%) | 10 (23%) | 8 (24%) | 5 (29%) | 2 (25%) | 4 (27%) | 39 (12) |

*Other = fungal skin infections, cellulitis, herpes stomatitis, congenital syphilis; **CAI = community-acquired infection

**Table 4. The estimated cost of hospitalization, investigation and treatment of paediatric community-acquired infections.**

| CAI episode | Number of episodes observed | Mean length of stay (days) | Overall cost per CAI episode | Cost per CAI episode |
|---|---|---|---|---|
| Community acquired pneumonia | 197 | 7.0 | R 4,021164 | R 20 412 |
| Diarrhoeal disease | 51 | 8.2 | R 1,223,932 | R 23 998 |
| Mixed infection | 43 | 9.3 | R 1,177,393 | R 27 381 |
| Blood-stream Infection | 33 | 9.7 | R 938,223 | R 28 431 |
| Meningitis | 17 | 6.6 | R 329,158 | R 19 382 |
| Urinary tract infection | 8 | 9.7 | R 227448 | R 28 431 |
| Other infections | 15 | 6.5 | R 285,622 | R 19 041 |
| **Total CAI episodes** | 364 | 7.7 | **R 8,202,941** | **R 22 535** |

*Cost analysis of the impact of CAI was performed from the healthcare provider perspective using the 2015 costs entered into the formula (for each of the subtypes): number of CAI events ×average length of stay for that subtype × unit cost per patient (unit cost per patient = R2916, including all laboratory investigations, radiology and pharmaceutical cost).* mixed = more than 1 CAI type; Other = fungal skin infections, cellulitis, herpes stomatitis, congenital syphilis

The most prevalent bloodstream pathogens in this study population were *E. coli*, pathogenic streptococci and non-typhoidal salmonella. This profile differed from a previous study (2009–2014) at the same hospital which noted that *K. pneumoniae*, *S. aureus* and *E. coli* were the most frequent paediatric bacteremia pathogens. The high rate of pre-transfer antibiotics (46%) in this cohort may have reduced the observed blood culture yield. Blood culture contamination rates in this study (7%) approximated that reported at another tertiary hospital in our province (6.6%), but were substantially lower than that reported from a Nigerian hospital [19]. The major blood culture contaminants in the studies were coagulase negative staphylococci, highlighting the importance of proper skin preparation before sampling of blood is done.

There is a need to produce local guidelines that emphasize judicious use of laboratory investigations to most effectively utilize the limited healthcare resources available in our middle-income country setting. Cautious use of laboratory investigations and antimicrobials will help reduce the overall cost of hospital management for CAI. In this study, the community acquired pneumonia and GE were leading contributors to CAI cost (49% and 15% respectively), with overall costs estimated at 8 million ZAR (approximately 500 000 USD).

Empiric antimicrobial prescriptions for CAI episodes at our institution were generally compliant with national guidelines (90%), including stewardship best practices such as prompt stopping of antibiotics, and de-escalation or escalation of antibiotics based on clinical picture and laboratory findings. An area for improved antimicrobial stewardship practice identified in this study is prescription of antibiotics to children with GE and no risk factors for bacterial infection. Furthermore, frequent review of empiric antibiotic guidelines (at least annually) is a clear priority given the high rates of antibiotic resistant pathogens encountered in paediatric CAI at our institution.

Limitations of this study include lack of generalizability (inclusion of data from a single, tertiary paediatric center), lack of data on the routine immunization status of the cohort, and the possible effect of pre-hospital administration of antibiotics on culture yield. Strengths of this study are the inclusion of detailed clinical and demographic data, access to advanced diagnostic microbiology and virology services, and inclusion of data seldom reported in studies of paediatric CAI from LMIC such as appropriateness of antimicrobial therapy, laboratory diagnostic use and cost of hospital management. Future studies should incorporate evaluation of antimicrobial and diagnostic test use for management of paediatric CAI, especially from LMIC where these data are seldom reported.

## Conclusion

CAI was the most common reason for paediatric hospitalization, driving antimicrobial use, hospital and laboratory costs. Empiric antibiotic prescription practice for CAI was generally compliant with national guidelines. Improved diagnostic stewardship is needed to reduce culture contamination rates, improve pathogen yield, and decrease unnecessary antimicrobial use. Clinical outcomes of paediatric CAI episodes were generally favourable.

## Supporting information

**S1 Appendix. Integrated management of childhood illness.**
(PDF)

**S2 Appendix. Standard treatment guidelines and essential drug list 2013.**
(PDF)

**S3 Appendix. Minimal dataset.**
(XLSX)

## Acknowledgments

The authors thank the patients, their parents and the clinical staff of the Tygerberg Hospital general paediatric ward G7.

## Author Contributions

**Conceptualization:** Angela Dramowski.

**Data curation:** Lydia Mapala, Angela Dramowski.

**Formal analysis:** Lydia Mapala.

**Funding acquisition:** Angela Dramowski.

**Investigation:** Lydia Mapala, Adrie Bekker, Angela Dramowski.

**Methodology:** Lydia Mapala, Adrie Bekker.

**Resources:** Lydia Mapala.

**Software:** Lydia Mapala, Adrie Bekker, Angela Dramowski.

**Supervision:** Adrie Bekker, Angela Dramowski.

**Validation:** Angela Dramowski.

**Visualization:** Adrie Bekker, Angela Dramowski.

**Writing – original draft:** Lydia Mapala.

**Writing – review & editing:** Lydia Mapala, Adrie Bekker, Angela Dramowski.

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
