## [Decision Letter · Decision Letter 0]

30 Mar 2022

PONE-D-21-14184Evaluating the appropriateness of laboratory testing and antimicrobial use in South African children hospitalized for community-acquired infectionsPLOS ONE

Dear Dr. Mapala,

Thank you for submitting your manuscript to PLOS ONE. After careful consideration, we feel that it has merit but does not fully meet PLOS ONE’s publication criteria as it currently stands. Therefore, we invite you to submit a revised version of the manuscript that addresses the points raised during the review process.

The manuscript has been evaluated by three reviewers, and their comments are available below.

The reviewers have raised a number of concerns that need attention. Please disregard Reviewer 2's comments on the manuscript title, but please consider the content of the remaining comments in your revisions.

Could you please revise the manuscript to carefully address the concerns raised?

We look forward to receiving your revised manuscript.

Kind regards,

Sebastian Shepherd

Associate Editor

PLOS ONE

Journal Requirements:

3. Thank you for providing the date(s) when patient medical information was initially recorded. Please also include the date(s) on which your research team accessed the databases/records to obtain the retrospective data used in your study.

5. Thank you for stating the following financial disclosure: "AD is supported by a NIH Fogarty Emerging Global Leader Award K43 TW010682 and received funding for the laboratory work from the SAMRC through a Self-initiated Research (SIR) Grant."

7. We note you have included a table to which you do not refer in the text of your manuscript. Please ensure that you refer to Table 4 in your text; if accepted, production will need this reference to link the reader to the Table.

Reviewers' comments:

Reviewer's Responses to Questions

**Comments to the Author**

1. Is the manuscript technically sound, and do the data support the conclusions?

Reviewer #1: Yes

Reviewer #2: No

Reviewer #3: Yes

2. Has the statistical analysis been performed appropriately and rigorously? 

Reviewer #1: Yes

Reviewer #2: N/A

Reviewer #3: Yes

3. Have the authors made all data underlying the findings in their manuscript fully available?

Reviewer #1: Yes

Reviewer #2: No

Reviewer #3: Yes

4. Is the manuscript presented in an intelligible fashion and written in standard English?

Reviewer #1: Yes

Reviewer #2: No

Reviewer #3: Yes

5. Review Comments to the Author

Reviewer #1: the manuscript is technically sound and relevant to the field.

it can inform broader studies and highlight needs in similar settings.

data analysis well structured.

manuscript written in intelligible fashion.

Reviewer #2: Following are comments for the authors,

1 The title of the study is not meaningful or fascinating, it should be like 'Evaluating the appropriateness of laboratory testing and antimicrobial use among South hospitalized children with community acquired infections. I wonder how a single center study can represent whole country so avoid using 'South Africa'.

1. Second paragraph of the introduction section need to update to provide per year episode of infection in precise way.

2. introduction section is not written well and lack flow to provide study rationale, coordination between infection type, appropriate testing, appropriate antibiotics regimen in sequence wise.

3. introduction section, paragraph 2 only stated that ceftriaxone is the appropriate antibiotic for CAP, this section should enlist all the antibiotics suitable for treating CAP.

4. No information about appropriate antibiotics for gastroenteritis, UTI were mentioned.

5. Introduction section must have, global, regional burden of CAIs among children and availability of testing facilities.

6. Inclusion exclusion criteria not included in this section. Study population characteristics like age also not provided in the method section.

7. Data presentation is not according to the standard

8. Several English grammar were clear in the main text of the paper.

9. References were not properly cited.

Reviewer #3: Dear Editor, I thank you for asking to review this submission by authors from pediatric hospital in South Africa. The study is a secondary data analysis and shows importance of diagnostic stewardship for the appropriate use of antimicrobial agents for treatment of community acquired infections. Authors performed a cost analysis as well. Title and objectives are in the synchrony. Authors described common community acquired infections, causing pathogens and related cost in pediatric patients. Tables are well designed.

One comment:

Authors mentioned that the most common pathogens causing UTIs are E. coli and K. pneumoniae with ESBL pattern. Authors should clarify what admission criteria are used to categorize them as community acquired infections.

No further comments are indicated.

6. PLOS authors have the option to publish the peer review history of their article (what does this mean?). If published, this will include your full peer review and any attached files.

Reviewer #1: **Yes: **Gilberto Luciano Lucas

Reviewer #2: **Yes: **Zia Ul Mustafa

Reviewer #3: **Yes: **Olga Perovic

---

## [Author Response · Author response to Decision Letter 0]

12 Jun 2022

Reviewer #1:

The manuscript is technically sound and relevant to the field. It can inform broader studies and highlight needs in similar settings, data analysis well structured. Manuscript written in intelligible fashion. Thank you

Reviewer #2:

The title of the study is not meaningful or fascinating, it should be like 'Evaluating the appropriateness of laboratory testing and antimicrobial use among South hospitalized children with community acquired infections. I wonder how a single center study can represent whole country so avoid using 'South Africa'. 

As per the associate Editors advice, we have opted to keep the original title of the manuscript.

 Second paragraph of the introduction section need to update to provide per year episode of infection in precise way. 

This has been improved to add that there has been a decrease in the incidence of Community acquired pneumonia globally by 30% between 2010 to 2015. 

Introduction section is not written well and lack flow to provide study rationale, coordination between infection type, appropriate testing, appropriate antibiotics regimen in sequence wise.

The introduction section has been edited to hopefully provide better flow and rational for the study.

Introduction section, paragraph 2 only stated that ceftriaxone is the appropriate antibiotic for CAP, this section should enlist all the antibiotics suitable for treating CAP. 

This has been updated to state that; For acutely ill children requiring transfer from primary care to hospital, the Integrated Management of Childhood Illness guidelines, and South African Essential Drug List (EDL) recommends a pre-transfer dose of intravenous or intramuscular third-generation cephalosporin e.g., ceftriaxone Oral amoxicillin is first line therapy for non-severe pneumonia. Ampicillin gentamicin for infants less than 3 months with severe pneumonia and Ceftriaxone for others with severe pneumonia.

No information about appropriate antibiotics for gastroenteritis, UTI were mentioned. 

This has been updated to reflect that Antibiotics were not recommended routinely for diarrhoeal disease as most is due to viruses. In cases such as severe malnutrition ampicillin and gentamicin was the recommended therapy ,infants less than 28 days, ampicillin gentamicin preferred. In suspected dysentery among Ceftriaxone or Ciprofloxacillin was the recommended treatment.

For Urinary tract infection, children more than 3 months old and not acutely ill, the recommended therapy was oral Amoxicillin/Clavulanic acid. Intravenous Amoxicillin/clavulanic acid or ceftriaxone was for very ill children or infants less than 3 months old

Introduction section must have, global, regional burden of CAIs among children and availability of testing facilities.

We included what we could we find about the global, regional, and local trends in CAIs but unfortunately data on the pathogen spectrum, management and disease outcome is scarce. 

Inclusion exclusion criteria not included in this section. Study population characteristics like age also not provided in the method section. 

The manuscript has been amended to state the inclusion and exclusion criteria better.

Inclusion Criteria

All children 0-13 years admitted to the general paediatric ward over the 6-month study period were eligible for inclusion in this study if their primary indication for hospitalization was a CAI episode. 

Hospitalization that lasted at least 48 hours

Pathogens isolated from laboratory samples (urine, CSF, blood, stool, respiratory aspirates) collected ≤ 48hrs after admission were classified as CAI, in patients with no prior history of hospitalization in the preceding 30 days.

Exclusion criteria

Patients considered to have a healthcare-associated infection (readmission within <30 days of hospital discharge) and patients with a non-infectious indication for hospitalization were excluded from further analysis

Data presentation is not according to the standard

The data tables will be improved in the process of copy editing and layout. 

Several English grammar were clear in the main text of the paper. 

We have attempted to improve the grammar throughout the manuscript

References were not properly cited. 

We have updated these using the PLOS ONE style format

Reviewer #3: 

Authors mentioned that the most common pathogens causing UTIs are E. coli and K. pneumoniae with ESBL pattern. Authors should clarify what admission criteria are used to categorize them as community acquired infections. 

CAI episodes were classified as described in the hospital records as infection episodes with pathogens isolated from laboratory samples (urine, CSF, blood, stool, respiratory aspirates) that were collected ≤ 48hrs after admission in patients with no prior history of hospitalization in the preceding 30 days. This was included in the Methods section of the manuscript.

Editors comments 

Please ensure that your manuscript meets PLOS ONE's style requirements, including those for file naming 

Manuscript was edited to meet PLOS ONE's style requirements

Please ensure that you include a title page within your main document. Could you therefore please include the title page into the beginning of your manuscript file itself, listing all authors and affiliations. 

Thank you, we have inserted a title page.

Thank you for providing the date(s) when patient medical information was initially recorded. Please also include the date(s) on which your research team accessed the databases/records to obtain the retrospective data used in your study. 

Manuscript has been amended to reflect that the database was accessed again from August 2017 for the present study

We note that the grant information you provided in the ‘Funding Information’ and ‘Financial Disclosure’ sections do not match. When you resubmit, please ensure that you provide the correct grant numbers for the awards you received for your study in the ‘Funding Information’ section.

Thank you, we have corrected the online sections mentioning the NIH funding support for Prof Dramowski.

Thank you for stating the following financial disclosure: "AD is supported by a NIH Fogarty Emerging Global Leader Award K43 TW010682 and received funding for the laboratory work from the SAMRC through a Self-initiated Research (SIR) Grant." Please state what role the funders took in the study. If the funders had no role, please state: "The funders had no role in study design, data collection and analysis, decision to publish, or preparation of the manuscript." 

If this statement is not correct you must amend it as needed. Please include this amended Role of Funder statement in your cover letter; we will change the online submission form on your behalf.

 In your Data Availability statement, you have not specified where the minimal data set underlying the results described in your manuscript can be found. PLOS defines a study's minimal data set as the underlying data used to reach the conclusions drawn in the manuscript and any additional data required to replicate the reported study findings in their entirety. All PLOS journals require that the minimal data set be made fully available. 

Data has been made available as Supporting information files, S3 Appendix: Minimal dataset

We note you have included a table to which you do not refer in the text of your manuscript. Please ensure that you refer to Table 4 in your text; if accepted, production will need this reference to link the reader to the Table. 

The manuscript was amended to reflect a reference to Table 4.

Please include captions for your Supporting Information files at the end of your manuscript, and update any in-text citations to match accordingly. Please see our Supporting Information guidelines for more information. 

This has been updated to include supporting files

S1 Appendix: Integrated Management of Childhood Illness

S2 Appendix: Standard Treatment Guidelines and Essential Drug List 2013

Thank you, we have edited the references. We have rechecked the references and have not identified any retracted papers.

---

## [Editor Report · Decision Letter 1]

14 Jul 2022

Evaluating the appropriateness of laboratory testing and antimicrobial use in South African children hospitalized for community-acquired infections

PONE-D-21-14184R1

Dear Dr. Mapala Lydia,

We’re pleased to inform you that your manuscript has been judged scientifically suitable for publication and will be formally accepted for publication once it meets all outstanding technical requirements.

Kind regards,

Olga Perovic

Guest Editor

PLOS ONE
---

## [Editor Report · Acceptance letter]

20 Jul 2022

PONE-D-21-14184R1 

Evaluating the appropriateness of laboratory testing and antimicrobial use in South African children hospitalized for community-acquired infections 

Dear Dr. Mapala:

I'm pleased to inform you that your manuscript has been deemed suitable for publication in PLOS ONE. Congratulations! Your manuscript is now with our production department. 

Kind regards, 

on behalf of

Professor Olga Perovic 

Guest Editor

PLOS ONE